# The Performances of Three Commercially Available Assays for the Detection of SARS-CoV-2 Antibodies at Different Time Points Following SARS-CoV-2 Infection

**DOI:** 10.3390/v14102196

**Published:** 2022-10-05

**Authors:** Heidi Syre, Marius Eduardo Brå Obreque, Ingvild Dalen, Åse Garløv Riis, Åse Berg, Iren Høyland Löhr, Jon Sundal, Lars Kåre Kleppe, May Sissel Vadla, Ole Bernt Lenning, Jan Stefan Olofsson, Kristin Greve-Isdahl Mohn, Camilla Tøndel, Bjørn Blomberg, Mai Chi Trieu, Nina Langeland, Rebecca Jane Cox

**Affiliations:** 1Department of Medical Microbiology, Stavanger University Hospital, 4068 Stavanger, Norway; 2Department of Research, Section of Biostatistics, Stavanger University Hospital, 4068 Stavanger, Norway; 3Department of Medicine, Stavanger University Hospital, 4068 Stavanger, Norway; 4Municipality of Randaberg, 4070 Randaberg, Norway; 5Faculty of Health Science, University of Stavanger, 4036 Stavanger, Norway; 6Department of Research, Stavanger University Hospital, 4068 Stavanger, Norway; 7Influenza Centre, Department of Clinical Science, University of Bergen, 5007 Bergen, Norway; 8Department of Clinical Science, University of Bergen, 5007 Bergen, Norway; 9Department of Medicine, Haukeland University Hospital, 5021 Bergen, Norway; 10Department of Microbiology, Haukeland University Hospital, 5021 Bergen, Norway

**Keywords:** SARS-CoV-2, antibody assay, spike, receptor binding domain, nucleocapsid

## Abstract

The aim of this study was to evaluate the performances of three commercially available antibody assays for the detection of severe acute respiratory syndrome coronavirus 2 (SARS-CoV-2) antibodies at different time points following SARS-CoV-2 infection. Sera from 536 cases, including 207 SARS-CoV-2 PCR positive, were tested for SARS-CoV-2 antibodies with the Wantai receptor binding domain (RBD) total antibody assay, Liaison S1/S2 IgG assay and Alinity i nucleocapsid IgG assay and compared to a two-step reference ELISA (SARS-CoV-2 RBD IgG and SARS-CoV-2 spike IgG). Diagnostic sensitivity, specificity, predictive values and Cohen’s kappa were calculated for the commercial assays. The assay’s sensitivities varied greatly, from 68.7% to 95.3%, but the specificities remained high (96.9–99.1%). The three tests showed good performances in sera sampled 31 to 60 days after PCR positivity compared to the reference ELISA. The total antibody test performed better than the IgG tests the first 30 days and the nucleocapsid IgG test showed reduced sensitivity two months or more after PCR positivity. Hence, the test performances at different time points should be taken into consideration in clinical practice and epidemiological studies. Spike or RBD IgG tests are preferable in sera sampled more than two months following SARS-CoV-2 infection.

## 1. Introduction

The coronavirus disease 2019 (COVID-19) pandemic has affected hundreds of millions of people and has claimed millions of lives. High quality diagnostic tests are important to limit the spread of severe acute respiratory syndrome coronavirus 2 (SARS-CoV-2), and during the pandemic, multiple SARS-CoV-2 diagnostic tests have been developed. Nucleic acid amplification tests (NAATs) detect viral RNA and are the preferred diagnostic tools for the detection of acute SARS-CoV-2 infection. The availability and costs may limit the use of NAATs, and SARS-CoV-2 antigen tests may be alternative tests for the detection of acute infection.

Antibody tests are indirect tests that detect the host’s immune response to SARS-CoV-2, induced either through natural infection or vaccination. The tests detect responses against the internal nucleocapsid protein or the external spike protein, which constitute the S1 and S2 subunits including the receptor-binding domain (RBD) of the S1 subunit. The detection of nucleocapsid antibodies indicates a resolving or past infection, whereas spike antibodies cannot distinguish between past infection and/or vaccination. Around 30% of SARS-CoV-2-infected persons seroconvert the first week following symptom onset, increasing to approximately 70% and 90% by the second and third week, respectively [1]. Persons with asymptomatic or mild disease normally have lower antibody levels than those with pronounced symptoms [2,3], reflecting higher viral replication rates and immune activation in patients with severe disease. IgM antibodies are usually detected before IgG but decline more rapidly. IgG persists for more than 12 months [4,5], and in some cases for 18 months [6,7] after past COVID-19 infection. The precise duration for the detection of IgG is unknown, but it is assumed that the antibodies give some level of immunity for at least a year [5]. The degree and duration of protection after infection compared to vaccination is unknown.

Antibody tests are especially suitable for monitoring viral spread in the community for epidemiological surveillance, since they detect infection in asymptomatic persons and thus have increased detection rates compared to NAATs [8,9]. Antibody tests are also useful in the diagnosis of persons with clinical symptoms of COVID-19 of longer duration or with complications, such as post-COVID-19 syndrome, multisystem inflammatory syndrome in children, myocarditis in young adults or other complications of COVID-19, where the NAATs have decreased sensitivity [10,11]. Furthermore, they may be of value when NAATs are not available or when it is difficult to take an airway sample of good quality. Antibody tests may also be used to differentiate past infection from vaccination by using tests that detect antibodies from different protein targets, and to identify whether vaccination was successful for inducing antibody responses in immunocompromised individuals [12].

Hundreds of tests have been developed for the detection of SARS-CoV-2 antibodies, including commercially available high-throughput automated tests for the use in clinical laboratories. However, the performances several months after symptom onset are not well described [1,2,13]. Few studies have evaluated the tests in clinically relevant study populations including contacts of PCR-positive patients [14], health care workers (HCWs) at high risk of SARS-CoV-2 infection [15], or in patients with mild symptoms or asymptomatic individuals. The aim of this study was to evaluate the performances of three commercially available antibody assays in PCR positive patients and healthy HCWs at high risk of SARS-CoV-2 infection, using a two-step ELISA as reference test. Sera from PCR-positive persons were collected at different time points following PCR positivity to investigate the duration of antibody positivity.

## 2. Materials and Methods

### 2.1. Patients and Samples

From March 2020 to June 2021, a prospective multicenter cohort study of HCWs with exposure to COVID-19-infected patients, and on patients suspected to be infected with SARS-CoV-2, was conducted in western Norway. The study was approved by the Regional Committee for Medical and Health Research Ethics in Norway (reference no. 118664), and with written informed consent from all study participants. Sampling was performed at inclusion, and at 4–8 weeks, 6 months and 1 year after inclusion. Serum samples were coded with a unique identification number, aliquoted, and stored at −80 °C upon analysis. Persons with symptoms of COVID-19 or with known exposure to SARS-CoV-2 were examined for SARS-CoV-2 RNA in naso- and/or oropharyngeal swab samples by real-time PCR.

Serum samples from 536 cases in the multicenter study were retrospectively included in the current study. Thus, cases could be negative, positive or in a grey zone area for SARS-CoV-2 antibodies in the reference ELISA test. The study population included all study cases (n = 368) from one study site and a random selection of PCR positive persons (n = 168) from the other study sites to increase the sample size. Only one serum sample from each case was included. The cases were stratified into SARS-CoV-2 PCR positive, PCR negative and PCR not performed. In addition, from 46 PCR-confirmed SARS-CoV-2-infected persons, sera collected at two to five different time points following PCR positivity were included to evaluate the performance of the assays at different time points after infection.

### 2.2. Two-Step Reference ELISA

The reference test for the detection of SARS-CoV-2 antibodies was a two-step enzyme-linked immunosorbent assay (ELISA), including a screening ELISA for the detection of anti-RBD reactive samples followed by a confirmatory RBD IgG and spike IgG ELISA. The antigens were produced in house in batches using the constructs based on the genomic sequence of the Wuhan-Hu-1 virus and by modifications of the method used by Amanat and colleagues [16]. The antigens were quality controlled and tested in the ELISA with a monoclonal antibody (CR3022) to ensure the antigenicity.

The reference ELISA was performed as previously described [16], with minor modifications [9]. Briefly, sera were tested in duplicates in 96-well plates to detect total immunoglobulins binding to RBD using 3,3′,5,5′-tetramethylbenzidine (BD Biosciences, Franklin Lakes, NJ, USA). A negative control panel of pre-pandemic sera (n = 128) and a positive control panel of PCR-confirmed COVID-19 patient sera (n = 43) were used to define the negative and positive cutoffs, respectively, based upon the optical density (OD) at 450/620 nm. Positive (OD > 0.708) or indeterminate (OD > 0.430) sera for anti-RBD were further analyzed using RBD IgG and a confirmatory spike IgG. Samples positive for spike IgG were defined as endpoint titers above three standard deviations of the mean of pre-pandemic negative control sera pool (titer > 485) [9]. A pre-pandemic sera pool, a hospitalized patient serum, and the human monoclonal antibody reactive to both SARS-CoV-1 and 2 (CR3022) were used as controls.

### 2.3. Commercially Available Serological Assays

Serum samples were analyzed for RBD total antibodies by the Wantai SARS-CoV-2 Ab ELISA (Beijing Wantai Biological Pharmacy Enterprise Co., Ltd., Beijing, China) in the DYNEX DS2 system (Dynex Technologies, Chantilly, VA, USA), for S1 and S2 IgG by the LIAISON SARS-CoV-2 S1/S2 IgG assay (DiaSorin, Saluggia, Italy) in the Liaison XL analyzer (DiaSorin), and for nucleocapsid IgG by the SARS-CoV-2 IgG (Abbott Laboratories, Abbott Park, IL, USA) in the Alinity i system (Abbott), according to the manufacturer’s instructions. The assays chosen were based on the platforms available at our routine microbiological laboratory. Laboratory personnel were blinded for test results in SARS-CoV-2 PCR and reference ELISA during analysis and interpretation. Characteristics of the commercially available assays are listed in Table 1. Test results in the grey zone (intermediate) area were interpreted as negative.

### 2.4. SARS-CoV-2 PCR

Naso/oropharyngeal swab samples (FLOQSwab; Copan, Brescia, Italy) for detection of SARS-CoV-2 RNA were collected in 3 mL sterile universal transport medium (UTM; Copan) and stored at 4 °C for up to four days prior to testing. Viral RNA was extracted from 200 µL sample using the RNAdvance Viral Reagent Kit (Beckman Coulter Inc., Brea, CA, USA) in a Biomek i7 Automated Workstation (Beckman Coulter Inc.), according to the manufacturer’s instructions. Real-time reverse transcriptase PCR was performed in a QuantStudio 6 system (Applied Biosystems, Beijing, China) targeting the envelope (*E*) gene specific to the Sarbecovirus [17]. The limit of detection was 20 viral RNA genome copies/mL. The results were reported as positive (Ct values ≤ 40) or negative (Ct values > 40).

### 2.5. Statistical Analysis

Diagnostic categories were compared between the serological assays by cross-tabulation. Sensitivities and specificities of the commercial serological assays were estimated by comparison with the reference two-step ELISA and were presented with 95% Wilson confidence intervals (CI). Likewise, total agreement between commercial serological assays and ELISA was presented with Wilson CI. Chance-adjusted agreement was estimated as Cohen’s kappa (K) and reported with analytical CI. Cohen’s kappa was interpreted as follows: <0.20: poor agreement; 0.21–0.40: fair agreement; 0.41–0.60: moderate agreement; 0.61–0.80: good agreement; 0.81–1.00: very good agreement. Positive predictive values (PPVs) and negative predictive values (NPVs) were estimated assuming seroprevalence values of 1, 5, 20 and 50%. Confidence intervals for predictive values were bootstrapped percentile intervals based on B = 1000 resamples. All statistical analyses were performed in Stata v. 16, using function kapci for kappa CI. Plots were created in IBM SPSS Statistics for Windows, Version 26.0 (IBM Corp, released 2019, Armonk, NY, USA). *p* values < 0.05 were considered statistically significant.

## 3. Results

### 3.1. Test Performances for Commercial Serological Assays

Of the 536 cases included in the study, 207 were SARS-CoV-2 PCR positive, 15 were PCR negative and 314 had no PCR test due to the absence of COVID-19 symptoms or negative contact history. Time from PCR positivity to collection of sera was available for 195 cases. The mean age was 45 years, and 79% of the cases were women. Characteristics of the study population are shown in Table 2.

In total, 211 cases were positive in the reference ELISA, of which 194 (91.4%) were PCR positive. Sensitivity, specificity, overall agreement, predictive values and Cohen’s kappa for each of the commercial assays using the two-step ELISA as reference are presented in Table 3. The highest sensitivity was obtained by the Wantai RBD total antibody assay (95.3%) and the lowest by the Alinity i nucleocapsid IgG assay (68.7%). The specificities ranged from 96.9% to 99.1%. The agreement was good for Alinity i nucleocapsid IgG assay and very good for Wantai RBD total antibody and Liaison S1/S2 IgG assays. Positive predictive values at a prevalence of 50% were 96.9% or higher for all three tests. The corresponding negative predictive values were 95.3% for the Wantai RBD total antibody assay, 86.3% for the Liaison S1/S2 IgG assay and 76.0% for the Alinity i nucleocapsid IgG assay.

The sensitivities for different categories of sampling times following PCR positivity for 195 cases are shown in Table 4. For the Wantai RBD total antibody assay, the sensitivities remained high (93.3–100%) at all time points. For the Liaison S1/S2 IgG and Alinity i nucleocapsid IgG assays, the sensitivities increased from 65.7% and 74.3%, respectively, at days 1–30, to their maximum sensitivities of 93.4% and 96.1%, respectively, at days 31–60. The Alinity i nucleocapsid IgG assay showed the most pronounced decrease in sensitivities two months or more following PCR positivity, and the sensitivity at days 181–325 was 37.2%.

Table 5 shows the discrepant test results between the two-step reference ELISA and the commercial assays. When compared to the reference ELISA, the Wantai RBD total antibody assay had ten false positive and ten false negative test results. Seven of the ten false positive and four of the ten false negative test results were identified in sera from PCR-confirmed positive cases. The corresponding numbers for the Liaison S1/S2 IgG assay were six (including serum from one PCR-confirmed positive case) and 33 (including sera from 26 PCR-confirmed positive cases), respectively. The Alinity i nucleocapsid IgG assay had three false positive (no sera from PCR-confirmed positive cases) and 66 false negative (including sera from 56 PCR-confirmed positive cases) test results. The between-method agreement for Wantai RBD total antibody and Liaison S1/S2 IgG was 92.2%, Cohens kappa 0.835 (0.787–0.882), for Wantai RBD total antibody and Alinity i nucleocapsid IgG 85.4%, Cohens kappa 0.679 (0.616–0.742), and for Liaison S1/S2 IgG and Alinity i nucleocapsid IgG 87.6%, Cohens kappa 0.713 (0.650–0.776).

### 3.2. Test Performances in Sera from Participants with Repeated Samplings

From 46 SARS-CoV-2 infected cases, sera were sampled at more than one time point following PCR positivity. For 13 cases, sampling was performed at two time points, for one case at three time points, for five cases at four time points and for 27 cases at five time points. Figure 1 shows the percent of cases with positive test results in the three commercial antibody assays at different time points following PCR positivity. At 0–2 weeks, the Wantai RBD total antibody assay showed the highest positivity rate. In weeks 3 through 17–21, the Wantai RBD total antibody and Alinity i nucleocapsid IgG assays showed >90% positivity rates, and the Liaison S1/S2 IgG assay showed positivity rates ranging from 80 to 86%. Between weeks 17–21 and 37–38, most of the cases with repeated samples were vaccinated, resulting in 100% positivity rates in Wantai RBD total antibody and Liaison S1/S2 IgG assays in weeks 37–38. For Alinity i the detection rate of nucleocapsid IgG in weeks 37–38 was 45%.

## 4. Discussion

Detection of antibodies is important to assess the seroprevalence for SARS-CoV-2 in different population groups or geographical areas. We evaluated the performances of three commercially available antibody assays using a two-step *in house* ELISA as reference. The sensitivities of the commercial assays varied greatly, from 68.7 to 95.3%, but the specificities remained high (96.9–99.1%). Similar variability in test performances in SARS-CoV-2 antibody tests have previously been reported [1,18,19,20,21,22].

In this cohort, the Alinity i nucleocapsid IgG assay had the lowest sensitivity of the commercial tests. Alinity i detects antibodies against a different part of the virus than the reference test, which may explain parts of the low performance. However, 56 of the 66 (84.8%) sera with false negative test results in the Alinity i nucleocapsid IgG assay were from PCR-positive cases, which one would expect to be positive for SARS-CoV-2 antibodies. The high number of false negative test results combined with the low number of false positive test results (n = 3) in the Alinity i nucleocapsid IgG assay may indicate that the cutoff for positive test results may be too high. In contrast, the Wantai RBD total antibody assay had a more suitable cutoff, with ten false positive and ten false negative test results when compared to the reference ELISA. An alternative explanation for the low performance of Alinity i may be that nucleocapsid IgG has a shorter half-life compared to spike or RBD antibodies. This is supported by the fact that the majority (39 of 56) of the false negative tests in Alinity i were sampled 3 months or more following PCR positivity. The Wantai RBD total antibody assay had the highest sensitivity when compared against the reference ELISA. The RBD antigen used in the assay is the same as for the first step of the reference ELISA, which may contribute to a high test performance. However, several anti-RBD based tests have shown the same high performance [21,23,24], and a recent study has shown that the number of RBD-specific memory B cells remained unchanged, and the RBD antibodies showed increased potency 6 months after symptom onset, probably due to antigen persistence [25].

The time of sampling following PCR positivity impacted the test performances. For all three tests, the sensitivities were higher in the sera sampled one to two months compared to the first month after a positive PCR test. In sera taken within one month following PCR positivity, Liaison S1/S2 IgG had the lowest sensitivity (65.7%), and Wantai RBD total antibody assay had the highest sensitivity (97.1%), probably due to the IgM component in the assay. In sera sampled later than two months following PCR positivity, the performance of Alinity i was most affected by time of sampling, with a reduction in sensitivity from 96.1% to 37.2% in samples drawn at one to two months compared to six to ten months. This is in concordance with previous studies. Grandjean and colleagues [26] showed that 99% of the cases were positive for anti-spike and 75% were positive for anti-nucleocapsid six months following SARS-CoV-2 infection. The corresponding numbers in a Swedish COVID-19 cohort 13 months post infection were 97% and 36%, respectively [27]. The importance of the time point of sampling following PCR positivity was further demonstrated in our cohort of 46 persons with more than one sampling, where over half of the cases were nucleocapsid IgG negative more than nine months following PCR positivity.

The specificities for the three commercial tests were excellent and were in concordance with previous studies [1,19,20,21] with low levels of cross-reactivity of antibodies from past infections of seasonal coronavirus. However, most previous studies have evaluated the specificity in pre-pandemic sera and not in clinical settings. It is well known that tests that include an IgM component are vulnerable to cross-reactivity with non-SARS-CoV-2 circulating antibodies from past infections with other pathogens. One of the assays evaluated in this study is a total antibody test, but the high test performance of the Wantai RBD total antibody assay shows that the IgM component does not contribute to a high number of false positive test results in this cohort. This is in agreement with Harritshøy and colleagues [21] who showed that total antibody assays had the highest diagnostic accuracy for SARS-CoV-2 infection.

In a clinical setting, the prevalence of the disease is important when evaluating the performance of a diagnostic test and may have a major effect on the predictive values. The Alinity i nucleocapsid IgG assay had the highest PPV but the lowest NPV. With a prevalence of 5% and a likely prevalence in national surveys prior to the introduction of COVID-19 vaccines, 16 (13 to 19) antibody positive individuals would be missed per 1000 tested in the Alinity i assay, and 203 (0 to 367) individuals would obtain a false positive test result. Corresponding numbers for the same assay in a setting with a prevalence of 50%, a value considered possible in a cohort of contacts of COVID-19 patients, would be 240 (202 to 272) and 13 (0 to 30), respectively. Thus, it is important to perform a thorough evaluation of the performances of novel SARS-CoV-2 antibody tests to know their utility in different settings and cohorts.

The strength of the study is that it includes a considerably large cohort of SARS-CoV-2 PCR-positive cases as well as HCWs at high risk of contracting SARS-CoV-2 infection. Sera were sampled at different time points following PCR positivity, ranging from 1 to 325 days, to perform a more clinical relevant evaluation of the commercial assays. This is in contrast to most previous studies that have few PCR-positive cases included and that evaluate the test performances one to three months past COVID-19 infection, when the antibody levels are at their peaks. There are some weaknesses in the study. The reference test used is a two-step ELISA for the detection of RBD and spike IgG. A more suitable reference would be a virus neutralization assay to measure the relationship between the level of antibodies in the commercial assays and the protective immunity against SARS-CoV-2. The variation in the sensitivity of the three serological assays evaluated in this study may reflect that some assays are highly sensitive for the detection of antibodies, whereas other assays correlate better with the level of neutralizing antibodies. Furthermore, the test performances were not evaluated in sera from immunocompromised persons or from children and were not properly evaluated in sera from vaccinated individuals.

## 5. Conclusions

In conclusion, when comparing three commercially available assays for the detection of SARS-CoV-2 antibodies to a reference *in house* ELISA, there was a noteworthy variation in sensitivities, but the specificities were in general high. The assays were evaluated in sera from a population where the majority of cases were women and the average age was 45 years. All three tests performed well in sera sampled 31 to 60 days following PCR positivity when the ELISA was used as reference. The Wantai RBD total antibody test performed better than the IgG tests the first 30 days following PCR positivity, and especially, the nucleocapsid IgG test showed reduced sensitivity two months or more after PCR positivity. The sensitivity of the Wantai RBD total antibody remained high through all time points evaluated. In this study, the specificities remained high for up to ten months following infection. The test performances at different time points after symptom onset should be taken into consideration in clinical practice and epidemiological studies, and spike or RBD antibody tests should be preferred in sera sampled more than two months following SARS-CoV-2 infection.

## Figures and Tables

**Figure 1 viruses-14-02196-f001:**
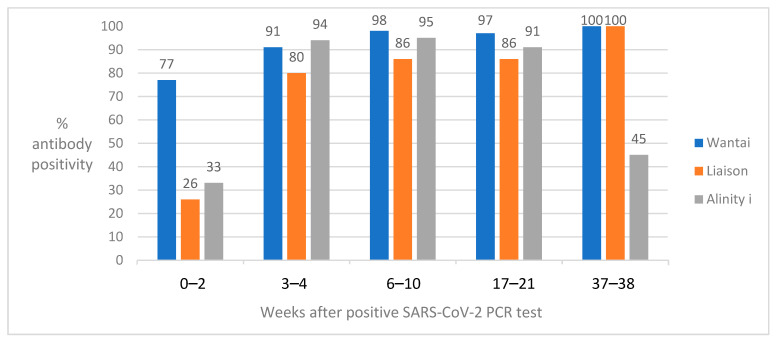
SARS-CoV-2 antibody positivity rates in the Wantai RBD total antibody assay, Liaison S1/S2 IgG assay and Alinity i nucleocapsid IgG assay by analyzing serum samples collected from 46 SARS-CoV-2 infected individuals at two or more time points for up to 38 weeks following PCR positivity. In weeks 37–38, most PCR-positive cases were SARS-CoV-2 vaccinated, resulting in 100% positivity rates in the Wantai RBD total antibody assay and Liaison S1/S2 IgG assay.

**Table 1 viruses-14-02196-t001:** Characteristics of the serological assays included in the study.

Characteristics	Two-Step ELISA	Wantai RBD Total Antibody	Liaison S1/S2 IgG	Alinity i Nucleocapsid IgG
Assay format	Two-step ELISA	Qualitative ELISA	Quantitative CLIA	Qualitative CMIA
Targeted viral antigen	RBD, Spike	RBD	S1/S2	Nucelocapsid
Immunoglobuline class	IgG	Total Ig	IgG	IgG
Measuring interval	Not given	Not given	3.8–400 AU/mL	Not given
Threshold for positive test	RBD IgG: OD > 0.708 Spike IgG: Titer > 485	Ratio ≥ 1.1	≥15 AU/mL	≥1.5 index (S/C)
Grey zone area	RBD IgG: OD 0.430–0.708 Spike IgG: - ^2^	0.9 ≤ Ratio <1.1	12–15 AU/mL	1.3–1.5 index (S/C) ^1^
Threshold for negative test	RBD IgG: OD < 0.430 Spike IgG: - ^2^	Ratio < 0.9	<12 AU/mL	<1.3 index (S/C)

^1^ Grey zone area established at the laboratory since not given by the manufacturer. ^2^ Grey zone area for spike IgG in the reference ELISA was not established. Abbreviations: ELISA: enzyme-linked immunosorbent assay, CLIA: chemiluminescent immunoassay, CMIA: chemiluminescent microparticle immunoassay, RBD: receptor-binding domain, S1/S2: subunit 1/subunit 2, Ig: immunoglobulin, AU: arbitrary unit, OD: optical density, S/C: sample control index ratio.

**Table 2 viruses-14-02196-t002:** Characteristics of the study population.

Characteristic	Total	SARS-CoV-2 PCR Positive	SARS-CoV-2 PCR Negative	SARS-CoV-2 PCR Not Performed
Number	536	207	15	314
Age, mean (range)	45.0 (14–89)	49.7 (20–89)	48.7 (28–76)	41.8 (14–75)
Male sex (%)	110 (21%)	64 (31%)	4 (27%)	42 (13%)

**Table 3 viruses-14-02196-t003:** Performances of the commercially available antibody tests when compared to the two-step reference ELISA.

Statistic Parameter	Number of Tests	Wantai RBD Total Antibody (95% CI)	Liaison S1/S2 IgG (95% CI)	Alinity i Nucleocapsid IgG (95% CI)
Sensitivity	211	95.30%	84.40%	68.70%
		(91.5–97.4)	(78.9–88.6)	(62.2–74.6)
Specificity	320	96.90%	98.20%	99.10%
		(94.4–98.3)	(96.0–99.2)	(97.3–99.7)
Agreement	536	96.30%	92.70%	87.10%
		(94.3–97.6)	(90.2–94.6)	(84.0–89.7)
Cohen’s kappa	536	0.922	0.844	0.715
		(0.888–0.955)	(0.797–0.891)	(0.655–0.776)
PPV				
-Prevalence 1%		23.8% (15.9–43.1)	31.6% (19.9–58.7)	42.9% (24.4–100)
-Prevalence 5%		62.0% (49.1–77.4)	70.6% (56.3–87.6)	79.7% (63.3–100)
-Prevalence 20%		88.6% (82.4–94.3)	92.0% (85.7–97.2)	94.9% (89.6–100)
-Prevalence 50%		96.9% (95.0–98.7)	97.9% (96.0–99.3)	98.7% (97.0–100)
NPV				
-Prevalence 1%		100% (100–100)	99.8% (99.8–99.9)	99.7% (99.6–99.7)
-Prevalence 5%		99.7% (99.6–99.9)	99.2% (98.9–99.4)	98.4% (98.1–98.7)
-Prevalence 20%		98.8% (98.0–99.5)	96.2% (95.0–97.3)	92.7% (91.2–93.9)
-Prevalence 50%		95.3% (92.7–98.0)	86.3% (82.5–90.1)	76.0% (72.8–79.8)

CI for SE and SP are Wilson intervals; CI for PPV and NPV are bootstrapped percentile intervals based on B = 1000 resamples. Abbreviations: RBD: Receptor-binding domain, S1/S2: Subunit 1/subunit 2, Ig: Immunoglobulin, CI: Confidence interval, SE: Standard error, SP: Statistical power, PPV: Positive predictive value, NPV: Negative predictive value.

**Table 4 viruses-14-02196-t004:** Sensitivities for the commercial assays in relation to sampling time following PCR positivity using the two-step ELISA as reference.

Assay	Sensitivity (95% CI), Sampling 1–30 Days after Positive PCR Test, n = 36	Sensitivity (95% CI), Sampling 31–60 Days after Positive PCR Test, n = 76	Sensitivity (95% CI), Sampling 61–90 Days after Positive PCR Test, n = 15	Sensitivity (95% CI), Sampling 91–180 Days after Positive PCR Test, n = 25	Sensitivity (95% CI), Sampling 181–325 Days after Positive PCR Test, n = 43
Wantai RBD total antibody	97.1% (85.5–99.5)	98.7% (92.9–99.8)	93.3% (70.2–98.8)	100% (86.7–100)	97.7% (87.9–99.6)
Liaison S1/S2 IgG	65.7% (49.2–79.2)	93.4% (85.5–97.2)	86.7% (62.1–96.3)	88.0% (70.0–95.8)	90.7% (78.4–96.3)
Alinity i nucleocapsid IgG	74.3% (57.9–85.8)	96.1% (89.0–98.6)	66.7% (41.7–84.8)	52.0% (33.5–70.0)	37.2% (24.4–52.1)

Abbreviations: RBD: receptor-binding domain, S1/S2: subunit 1/subunit 2, Ig: immunoglobulin, CI: confidence interval.

**Table 5 viruses-14-02196-t005:** Discordant results in the commercial antibody assays when compared to the two-step reference ELISA.

Assay	Discordant Category Compared to Reference Test	Number of Samples	PCR Positive	PCR Negative	PCR Not Performed
Wantai RBD total antibody	False positive	10	7	0	3
	False negative	10 (2 grey zone)	4 (1 grey zone)	0	6 (1 grey zone)
Liaison S1/S2 IgG	False positive	6	1	4	1
	False negative	33 (3 grey zone)	26 (3 grey zone)	0	7
Alinity i nucleocapsid IgG	False positive	3	0	3	0
	False negative	66 (4 grey zone)	56 (2 grey zone)	2 (2 grey zone)	8

Abbreviations: RBD: receptor-binding domain, S1/S2: subunit 1/subunit 2, Ig: immunoglobulin.

## Data Availability

Not applicable.

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
