# Peer review of "The Performances of Three Commercially Available Assays for the Detection of SARS-CoV-2 Antibodies at Different Time Points Following SARS-CoV-2 Infection"

_viruses, 2022, doi:10.3390/v14102196_

Round 1
Reviewer 1 Report
The present article uses three commercial serological kits for Sars CoV-2 detection and compares specificity, sensitivity, and nonspecificity between them. Some samples was employed to analyze the time of detection of every kit and and potendtial detection of each stretegy.The control was a two-step ELISA, but the antigem used for this test is not clarified, only cited (ref:17 and 9), especify the origem of the antigen for each test will turn the understending more easealy (in the text).In the table 1 is comented that the two-step ELISa use RBD and Spike,why the grey zone for RBD not is given?
In the figure 1,line 202 has two dots.
In the specifique parte of discution , line 268-274, does not make sense.The general ideia of the article is not correlate with detection and protection,the most weak point of the article is be limited intterested to countris, regions or companies
that use the same serological kits. The Originality / Novelty of the article is not high, but the large number of samples give to the article a good acurency in the resoults.
Author Response
Reviewer 1:
1: The control was a two-step ELISA, but the antigem used for this test is not clarified, only cited (ref:17 and 9), especify the origem of the antigen for each test will turn the understending more easealy (in the text).
Response: The antigens for the two-step ELISA were RBD and spike, as stated in lines 108-109 and in Table 1. We produced the antigens in batches in house using the constructs based on the genomic sequence of the Wuhan-Hu-1 virus,and by modifications of the method used by Amanat et al. Nat Med 26, 1033–1036, 2020. The RBD was produced in one large batch for all assays. The large spike protein was more difficult to purify in one large batch, and we produced a number of batches which were quality controlled and tested in the ELISA with a monoclonal antibody CR3022 to ensure the antigenicity of the spike, see graph below. The antigenicity of RBD was also tested with CR3022. We agree that the antigens need to be described in more details, and have added information on the origin of the antigens in line 110-113.
2: In the table 1 is commented that the two-step ELISA use RBD and Spike,why the grey zone for RBD not is given?
Response: The grey zone of RBD is in table 1 given to be OD 0.430 – 0.708, but a grey zone of spike was not established. This is added as footnote 2 in Table 1 (line 139-140) to make it more clear. The cutoffs for SARS-CoV-2 RBD ELISA were defined using pre-pandemic historical negative control (NC, n=128, gray in the figures below) serum samples and SARS-CoV-2 RT-PCR-confirmed positive control (PC, n=43, red in figures below) patients. The black lines indicate the median with interquartile range. Dotted lines are the negative cutoff at OD 0.430, calculated as 3 standard deviations above the mean of pre-pandemic NC and ensured 97.5% of NC were below the negative cutoff. The positive cutoff at OD 0.708, was calculated by 3 standard deviations below the mean of SARS-COV-2 RT-PCR-confirmed PC and ensured 97.5% of PC above the positive cutoff. For serum samples, the screening RBD ELISA results were interpreted as negative in samples with an OD < 0.430, positive if OD ≥ 0.708, and intermediate if 0.430 ≤ OD < 0.708. All samples with intermediate or positive anti-RBD screening OD values were further investigated for RBD IgG and a confirmatory spike IgG. The ELISA has previously been described in details by Trieu et al (reference 9).
3: In the figure 1, line 202 has two dots.
Responce: The extra dot is removed, thank you.
4: In the specifique parte of discution , line 268-274, does not make sense.
Response: Thank you. We agree that this may be difficult to read and have deleted line 272-274 (line 309-311 in revised version) and reference 28.
Reviewer 2 Report
The authors described the effectiveness of three commercially available antibody assays (Wantai SARS-CoV-2 Ab ELISA (Beijing Wantai Biological Pharmacy Enterprise Co., Ltd., China), the LIAISON SARS-CoV-2 S1/S2 IgG assay (DiaSorin, Italy) and SARS-CoV-2 IgG (Abbott Laboratories, USA) for detecting antibodies to SARS-CoV-2 at different time points after virus infection. The test values at different time points can be useful in clinical practice and epidemiological studies.
I have a couple of comments to improve the manuscript quality.
1) There are currently many commercially available test systems for detecting the presence of antibodies to SARS-CoV-2, including diagnostic tests from Roche, Siemens, Fisher Scientific and other. The authors should explain the algorithm for selecting specific test systems for immunoassays.
2) The reference test for the detection of antibodies to SARS-CoV-2 (two-step ELISA) should be described in detail in section 2. Materials and methods. In particular, the method of obtaining recombinant RBD and spike proteins, their characteristics or the company that produced these proteins should be specified.
3) Some interesting data presented in the tables are not discussed in the results section, neither in the discussion nor in the conclusion. This needs to be corrected.
Author Response
Reviewer 2:
1: There are currently many commercially available test systems for detecting the presence of antibodies to SARS-CoV-2, including diagnostic tests from Roche, Siemens, Fisher Scientific and other. The authors should explain the algorithm for selecting specific test systems for immunoassays.
Responce: In the study, we used the platforms available at our routine microbiological laboratory, i.e., an anti-RBD assay, an anti-spike assay and a total antibody assay. For a routine microbiological laboratory, it is useful to have assays that detect different parts of the virus to differentiate between vaccinated and previously infected persons, and in addition a total antibody test for rapid diagnostic of newly infected persons. We agree that this needs to be explained, and the algorithm for selecting the different test systems is described in line 133-134. Thank you.
2: The reference test for the detection of antibodies to SARS-CoV-2 (two-step ELISA) should be described in detail in section 2. Materials and methods. In particular, the method of obtaining recombinant RBD and spike proteins, their characteristics or the company that produced these proteins should be specified.
Responce: We agree that the reference ELISA needs more detailed description although the assay has previously been described in details by Trieu et al (reference 9), and we have added more details in line 114-125. Please also see response to reviewer 1 where we describe the two-step ELISA with RBD and spike.
3: Some interesting data presented in the tables are not discussed in the results section, neither in the discussion nor in the conclusion. This needs to be corrected.
Responce: Thank you for pointing this out. We have added the following to meet this recommendation:
-Table 2: More details on the study population in Results (line 172) and in Conclusions (line 320-322).
-Table 3: More details on the predictive values in Results (line 183-186). We believe that the rest of the results in Table 3 are sufficiently described in Results, Discussion and Conclusions.
-Table 4: More details on the sensitivities in relation to sampling time following PCR positivity in Results (line 192-197) and in Conclusions (line 326-327). We believe that the results in Table 4 are sufficiently discussed in the Discussion part.
-Table 5: More details on discordant results in Results (line 202-209) and in Discussion (line 247-251).